# Transcription of Autophagy Associated Gene Expression as Possible Predictors of a Colorectal Cancer Prognosis

**DOI:** 10.3390/biomedicines11020418

**Published:** 2023-01-31

**Authors:** Martyna Bednarczyk, Małgorzata Muc-Wierzgoń, Sylwia Dzięgielewska-Gęsiak, Edyta Fatyga, Dariusz Waniczek

**Affiliations:** 1Department of Hematology and Cancer Prevention, Medical University of Silesia in Katowice, 40-055 Katowice, Poland; 2Department of Preventive Medicine, Medical University of Silesia in Katowice, 40-055 Katowice, Poland; 3Department of Surgical Nursing and Propaedeutics of Surgery, Medical University of Silesia, 40-055 Katowice, Poland

**Keywords:** autophagy, *LAMP-2*, BECN1, PINK1, FOXO1 gene expression, RT-qPCR method, colorectal cancer

## Abstract

(1) Background: Autophagy plays a dual role in oncogenesis—it contributes to the growth of the tumor and can inhibit its development. The aim of this study was to assess changes in the transcriptional activity of *LAMP-2, BECN1, PINK1*, and *FOXO1* genes involved in the autophagy process in histopathologically confirmed adenocarcinoma sections of colorectal cancer: (2) Methods: A gene expression profile analysis was performed using HG-U133A and the RT-qPCR reaction. The transcriptional activity of genes was compared in sections of colorectal cancer in the four clinical stages (CSI-CSIV) concerning the control group; (3) Results: In CSI, the transcriptional activity of the *PINK1* gene is highest; in CS II, the *LAMP-2* gene is highest, while *FOXO1* increases gradually from CSI reaching a maximum in CSIII. There is no *BECN1* gene expression in colorectal cancer cells; (4) Conclusions: The observed differences in the mRNA concentration profile of autophagy-related genes in colon cancer specimens may indicate the role of autophagy in the pathogenesis of this cancer. Genes involved in autophagy may be diagnostic tools for colorectal cancer screening and personalized therapy in the future.

## 1. Introduction

Three morphologically and mechanistically distinct types of autophagy exist in cells: macroautophagy, microautophagy, and chaperone-associated autophagy, where macroautophagy is often referred to as autophagy [1,2].

Autophagy is an intracellular, catabolic degradation process in which aggregated proteins, damaged organelles, and pathogens are delivered to lysosomes where they are digested by lysosomal hydrolases [3,4,5,6]. On the one hand, cellular self-digestion is responsible for the survival of cells in unfavorable conditions, and on the other, it eliminates damaged proteins and organelles, protecting cells against neoplastic transformation [7,8]. Recent studies have shown that autophagy plays an essential role in the pathogenesis of many diseases, including cancer, heart disease, infections, neurodegenerative disorders, autoimmune diseases, metabolic diseases, viral infections especially SARS-Cov-2 and COVID-19, as well as aging and cell death [9,10]. It is considered to be a type II programmed cell death [11]. The potential ability of autophagy to modulate cell death has made this process one that is considered to have a therapeutic aim in cancer therapy [12,13]. In the early stages of the disease, autophagy has a protective effect by removing damaged proteins, thereby protecting cells from malignant transformation. However, in late-stage cancer, autophagy promotes tumor survival and growth by scavenging toxic free radicals or damaged proteins.

In addition, autophagy supports mitochondrial function and tumor metabolism, resulting in survival under stressed conditions [14,15,16]. At the same time, it acts as a cellular defense mechanism to prevent cancer cell death after treatment, contributes to cancer recurrence and metastasis, and inhibits anticancer therapy and cancer cell death [9,17,18].

Higher levels of autophagy have been observed in many types of cancer; e.g., the main autophagy regulatory gene Beclin1 is upregulated in colorectal cancer, gastric cancer, liver cancer, breast cancer, and cervical cancer, suggesting that autophagy may be involved in carcinogenesis and that Beclin1 plays a key role in tumor formation [19,20,21,22].

Autophagy primarily plays a cytoprotective role in cancer cells and may be induced by most cancer therapies, including radiotherapy and chemotherapy [23,24,25,26,27]. Given the pro-survival role of autophagy, its inhibition has been shown to increase treatment efficacy. Therefore, inhibition of autophagy is considered a potentially valuable therapeutic approach in combination with other anticancer therapies [16,28].

It is worth noting that there are drugs available in anticancer therapy that act on autophagy, both stimulating and inhibiting this process. Inhibition of autophagy is achieved using pharmacological inhibitors, such as chloroquine, hydroxychloroquine, bafilomycin A1, 3-methyladenine, or by silencing gene expression, such as *Atg5, BECN1*, *Atg10*, or *Atg12*. Among the anticancer drugs that stimulate autophagy, the most popular are *Bcl2* inhibitors and mTOR pathway inhibitors [29]. It is important to understand the molecular mechanisms of autophagy and to determine whether, in a given cancer, autophagy results in the survival or death of cancer cells under the influence of treatment.

The study aims to assess changes in expression patterns of genes involved in autophagy in colorectal cancer specimens (histopathological type—adenocarcinoma) in various stages of clinical cancer according to the ninth Union for International Cancer Control/American Joint Committee on Cancer (UICC/AJCC) classification. Colorectal cancer (CRC) is the third most common occurring cancer in men and the second most commonly occurring cancer in women. There were over 1.9 million new cases in 2020. In recent years, the global burden of CRC will increase by 60%, to over 2.2 million new cases and 1.1 million deaths by 2030 [30].

The premise for undertaking this study was the data from the literature on the research conducted so far, indicating the dual role of autophagy in the process of carcinogenesis. It has also been shown that the expression level of some autophagy-related genes is down-regulated or up-regulated in many types of cancer. Additionally, in the Affymetrix database, the number of autophagy-related genes has doubled in just a few months, indicating that autophagy has been a very active research area in recent years.

## 2. Materials and Methods

### 2.1. Ethics

The current study was conducted by the Declaration of Helsinki. The project was approved by the Bioethics Committee of the Medical University of Silesia in Katowice—decision No. KNW/0022/KB1/42/14 (20 June 2014).

The participants were informed in detail about the study and gave their written consent. Participation in the study was voluntary. Patients’ data have been encoded by the pseudonymization procedure, which means that personal data are processed in such a way that they cannot be assigned to a specific data subject without the use of an additional “key”.

### 2.2. Materials

The study group included 18 men and 21 women (age: 59–79 years; median age: 69 years). The study was carried out on specimens of the large intestine taken from patients with colorectal adenocarcinoma (histopathological type—adenocarcinoma) at four stages of clinical cancer according to the ninth classification of the Union for International Cancer Control/American Joint Committee on Cancer (UICC/AJCC). The patients were hospitalized in the Department of General, Colorectal, and Polytrauma Surgery and the Department of Surgical Nursing and Propaedeutics of Surgery, Faculty of Health Sciences in Katowice, Medical University of Silesia, Poland. Tissue samples were obtained during surgical resection of the colon affected by cancer, which was performed according to surgical treatment standards. The tissue samples were collected using classical surgical techniques without the use of electric or ultrasound instruments. The material obtained consisted of tumor tissue and/or healthy colon tissue. Healthy control tissue specimens were collected from an area 5 cm outside of the histologically negative margin, during the operation because of CC. All materials were taken by the same operational team to minimize mistakes. The cancer samples were obtained from the margin of the resected material to rule out the presence of necrotic tissue in the specimens. 

The qualification of patients for the study included diagnosed and histopathologically confirmed colorectal cancer, regardless of the clinical stage, normal BMI, over 18 years of age, and consent of the patient. Patients with previously diagnosed and treated cancer, coexisting systemic disease, mental illness, chronic viral hepatitis, as well as individuals taking (long-term) non-steroidal anti-inflammatory drugs, immunosuppressants, steroids, or other hormonal drugs, were not eligible. The clinicopathological characteristics of the study group are given in Table 1.

### 2.3. Methods

In total, 39 paired tissues (39 tumor and 39 control) were collected. The collected intestinal specimens were placed in sterile tubes containing RNA *later*^TM^ (Sigma) in the amount of 10 µL per 1 mg of tissue (200 µL RNA *later*^TM^ per 20 mg of tissue). They were stored for 24 h at 4 °C, then the sections were frozen at −80 °C until the next stage of analysis. Molecular studies were performed at the Faculty and Department of Molecular Biology of the Medical University of Silesia.

This study is a continuation of an earlier published study, in which a microarray analysis of the expression profile of genes involved in autophagy was performed in adenocarcinoma specimens and specimens taken from the surgical margin, histopathologically assessed as healthy intestine (control). Oligonucleotide microarrays (HGU 133A—Affymetrix) were used for the study [31].

Confirmation of the results of the comparative analysis of transcriptomes determined by the expression microarray technique was carried out using the RT-qPCR method, recognized as the "gold standard" in the validation of matrix experiments.

The number of mRNA copies of the analyzed genes was determined based on the analysis of the kinetics of the qRT-PCR reaction using the Engine OPTICON^TM^ sequence detector (MJ Research Inc., Watertown, MA, USA) and the QuantiTect^TM^ SYBRGreen RT-qPCR Kit (QIAGEN) reagents according to the manufacturer’s recommendations [32].

Reverse transcription (RT) and quantitative PCR were performed in one reaction mixture using kit reagents. The reaction mixture consisted of 2× QuantiTect SYBR Green RT-PCR Master Mix, forward and reverse primers 0.5 M each, RNA template, and pyrogen-free water. The total mixture was 10 μL. For all tested samples, RT-qPCR was also performed for the mRNA of the *β-actin* gene, which was the endogenous control. 

For each RT-qPCR reaction, a standard curve was constructed from which the Opticon™ DNA Engine Sequence Detector calculated the copy number of the test mRNA in the reaction mixture. The standard curve was determined from the RT-qPCR results of the quantitative template—a fragment of the *β-actin* gene (TaqMan^®^ DNA Template Reagents Kit and *β-actin* Control Reagent Kit—Applied Biosystems) in five different concentrations (from 400, 800, 2000, 4000 to 8000 copies *β-actin* of cDNA/µL). On the basis of the fluorescence curve recorded after each amplification cycle, the number of messenger RNA copies (messenger RNA) of each gene tested in the conversion to 1 µg of RNA was determined [33].

The reaction was performed using sequence-specific primer pairs for each gene tested (Sigma-Aldrich, St. Louis, MO, USA)—Table 2.

#### Statistical Analysis

The results obtained using the qRT-PCR technique were developed based on the Statistica 12.5 program (StatSoft, Tulsa, OK, USA). 

For each analyzed parameter, the most important elements of descriptive statistics were determined: mean, median, minimum and maximum value, standard deviation, and upper (75%) and lower (25%) quartiles. The normality of value distribution was checked using the Shapiro-Wilk test. Then, the results with a Gaussian distribution were analyzed with Student’s test. In case of Gaussian distribution, the results are expressed as mean and standard deviation (SD), while in the case of non-Gaussian distribution, the results are expressed as median and standard deviation.

## 3. Results

The transcriptional activity of genes involved in the autophagy process (*LAMP-2*, *BECN1, PINK1, FOXO1*) in colorectal cancer biopsies in four clinical stages of adenocarcinoma (CSI, CSII, CSIII, CSIV) was compared with that of controls (colon samples assessed as histopathologically normal). The results were normalized to the endogenous *b-Actin* control. Comparing the number of copies of *BECN1* mRNA/µg RNA and *LAMP-2* mRNA/µg RNA in the control and study samples, no statistically significant differences were found (one-way ANOVA), assuming that *p* < 0.05. The lack of statistically significant differences may be due to the small size of the study group in each stage of clinical advancement of adenocarcinoma.

The ANOVA test showed statistically significant differences only in the expression of *PINK1* (*p* = 0.044194). It was also noted that *PINK1* expression was statistically significantly higher in the CSI about the normal intestine (*p* = 0.004188) and to CSII (*p* = 0.024910)—Table 3. The post-hoc LSD test allowed us to observe a significant increase in the transcriptional activity of the *FOXO1* gene in CSIII compared to the control (*p* = 0.020474).

To determine the potential role of *BECN1* in the development of colorectal cancer, a RT-qPCR test was performed, which showed a decrease in *BECN1* expression in tumor samples compared to controls (Figure 1). In CSI, CSII, and CSIII, the level of expression is at a similar level, while in CSIV it was significantly reduced (Table 3).

The results of the RT-qPCR reaction for *LAMP-2* show the highest transcriptional activity for CSII.

In the results of the RT-qPCR reaction about *PINK1* gene—the highest transcriptional activity is observed for CSI and CSIII, *FOXO1* gene expression increases gradually from CSI, reaching a maximum in CSIII (Figure 2).

To sum up the results, no statistically significant differences were observed for *BECN1* and *LAMP-2* among the examined genes (LSD test).

The direction of changes in the expression of the examined genes determined by the qRT-PCR method in colorectal cancer and controls is presented in Table 4.

## 4. Discussion

The molecular mechanisms of colorectal cancer involved in the proliferation and apoptosis of colorectal cancer cells are poorly understood [34].

As previous studies have shown, autophagy has a double effect on carcinogenesis. It can inhibit or promote tumor growth. It plays a very important role in all stages of oncogenesis, including initiation, progression, promotion, metastasis, and treatment resistance. Therefore, it is believed that the modulation of autophagy may be a new therapeutic strategy in the treatment of cancer [35,36,37].

Among the genes associated with autophagy, the *BECN1* gene represents the first link between autophagy and cancer. *BECN1* is also considered a tumor suppressor gene. It takes part in the formation of the autophagosome already in the first phase of aerophagy [38,39,40]. In addition, it interacts with many positive and negative regulators of autophagy, which also affects the activity of the process and carcinogenesis [38]. It is involved in many signaling pathways related to the regulation of autophagy. Moreover, it is associated with poor prognosis and metastasis [41,42]. The results of our study are in line with studies by Hu et al., which also showed lower expression of *BECN1* in CRC compared to healthy intestine samples. In addition, Hu’s team showed that the lower expression of *BECN1* was associated with a poor prognosis in CRC, suggesting that *BECN1* may act as a suppressor and represent a new prognostic marker for patients with colorectal cancer. Data presented by Hu et al. indicate that *BECN1* deficiency in CRC samples has no effect on cancer cell growth but significantly increases their mobility and invasion. Interestingly, the role of *BECN1* in CRC samples varied in different studies, which may be related to the use of multiple research methods and indicate that *BECN1* may play a multidirectional role depending on the clinical stage of the disease [43]. There are indications that the Beclin1 gene plays an important role in tumor growth, although the mechanism is not fully understood. On the other hand, therapy targeted at the *BECN1* molecule can control tumor growth [44,45,46]. In a study by Park et al., in turn, a higher expression of *BECN1* in CRC samples was demonstrated; however, the patients had previously undergone chemotherapy (adjuvant chemotherapy based on 5-fluorouracil) [47]. There are conflicting reports regarding the relationship between *BECN1* expression and prognosis in human cancers [48,49,50,51,52,53]. The Park team showed that overexpression of *BECN1* in CRC was significantly associated with reduced survival in patients undergoing 5-FU adjuvant chemotherapy [47]. In contrast, other CRC studies have shown that high vs. low *BECN1* expression is associated with a poorer prognosis and shorter survival in patients treated with surgery alone, and high vs. low *BECN1* expression is associated with longer survival in patients after surgical removal of the tumor also undergoing chemotherapy based on 5-FU [52,53]. The Park study, as well as other reports, did not identify the association of *BECN1* with clinicopathological variables [47,54,55]. Therefore, the prognostic effect of *BECN1* may be due to its ability to confer chemoresistance to cancer cells. Inhibition of autophagy has been shown to sensitize colorectal cancer cells to 5-FU-induced apoptosis, which is consistent with the pro-survival role of autophagy [56,57]. However, further research is needed to determine the role of BECN1 as a predictive versus prognostic biomarker.

There have been many other studies suggesting that *BECN1* may be closely related to survival. Beclin1 is a central regulator of autophagy that interacts with multiple proteins (UVRAG, Atg14L, Bif-1, Rubicon, Ambra1, Survivin). The Beclin1 signaling pathway regulates both autophagy and apoptosis, the balance of which determines the effectiveness of the anti-cancer treatment. The BH3 domain of *Beclin1* is bound and inhibited by the anti-apoptotic proteins Bcl-2 or Bcl-XL, which may reduce its ability to induce autophagy [58,59].

Studies by Grasso et al. showed that combining various agents, such as selumetinib and cytarabine, with autophagy inhibitors (bafilomycin A1, chloroquine, or 3-methyladenine) enhanced the activity of selumetinib and cytarabine against colorectal cancer cells and leukemia cells, respectively [60,61]. Caspases, on the other hand, can split *Beclin1* during apoptosis, thus preventing its autophagic activity [62,63]. Beclin1 plays an anti-apoptotic role in response to chemotherapy, but the mechanism has not yet been fully described [64]. All of this data suggests that Beklina1 exerts extensive regulatory control over the processes of cancer cell death, which contributes to influencing the outcome of CRC patients.

Research by Park et al. suggests that *BECN1* expression predicts the efficacy of cytotoxic chemotherapy in CRC patients. In addition, the results of the study support the targeting of autophagy in vivo to enhance the efficacy of cytotoxic drugs used in the treatment of colorectal cancer. Currently, nearly 20 clinical trials have been registered with the National Cancer Institute investigating autophagy inhibition as a therapeutic strategy against a variety of human cancers. Hydroxychloroquine is most commonly used to inhibit autophagy; however, stronger and more specific autophagy inhibitors are needed [47,65].

Summarizing the results for *BECN1*, we showed reduced expression of *BECN1* in all stages of CRC compared to controls, with the lowest expression being observed in CSIV. Therefore, it has been suggested that autophagy may contribute to the development or progression of colorectal cancer [48,66]. In the future, BECN1 may be a target for molecular therapies. Beclin1 silencing can be lethal in tumors that depend on an enhanced autophagic cell response for survival.

*LAMP-2* (*Lysosome-associated membrane protein 2)* is a membrane protein present in lysosomes. It is the main regulator of chaperone-dependent autophagy (CMA) [67]. The main difference between CMA and other types of autophagy, such as macroautophagy and microautophagy, is the selective targeting and degradation of specific substrate proteins, without affecting organelles or neighboring proteins [68]. Previous research into the physiology or diseases associated with *LAMP-2* primarily focused on aging [69], renal hypertrophy [70], and neurodegenerative diseases such as Parkinson’s disease [71,72] and Alzheimer’s disease [73,74]. Recently, there have been reports revealing the pro-neoplastic role of *LAMP-2* [75,76]. According to the current state of knowledge, the increase in CMA activity, and thus the overexpression of the *LAMP-2* protein, causes the progression of cancers, including colorectal cancers. This finding indicates that the inhibition of chaperone-dependent autophagy may, in the future, become an effective treatment for highly advanced stages of cancer [77].

Wang et al. showed that *LAMP-2* is overexpressed in many types of cancer and is associated with a poor prognosis. In turn, inhibition of *LAMP-2* reverses macrophage activation, increasing tumor cytotoxicity and inhibiting cancer progression [78]. In another study, analogous results were obtained; colorectal cancer cells were characterized by increased expression of *LAMP-2* [79]. Numerous studies indicate that neoplastic transformation is associated with various structural changes in carbohydrates on the surface of cells, in particularly increased sialation and β1-6-linked branching of complex oligosaccharides related to asparagine (Asn) [80]. *LAMPs* are the main carriers of polylactosaminoglycans in various cells. The ability of *LAMP* to bind to the components of the extracellular matrix is inversely proportional to the degree of its glycosylation, and it is believed that increased branching of β1–6 molecules contributes to the increased metastasis potential by reducing adhesion [81]. Although increased glycosylation may play a role in reducing adhesion, *LAMP-2* likely carries oligosaccharide ligands that are recognized by adhesion molecules [82]. It was found that highly metastatic tumor cells adhere more effectively to endothelial cells compared to tumor cells with a low metastatic potential [83].

Additionally, a study by Futura et al. showed that in non-cancerous areas of samples taken from cancer patients, there was no significant increase in *LAMP-2* activity, which may suggest that there is no relationship between *LAMP-2* expression and cell proliferation. It can be assumed that *LAMP-2* is associated with cancer progression through a mechanism other than cell proliferation [79].

Lichter-Koneckie et al. provide potential clues to the function of *LAMP-2* in tumorigenesis. Indeed, the researchers found that the expression pattern of *LAMP-2* was specific to tissue type and cell type depending on the progress of differentiation. The researchers suggest that two distinct mechanisms at the transcriptional and post-transcriptional levels generate a variety of *LAMP-2* proteins that perform different developmental functions. However, to confirm this hypothesis, it is necessary to conduct additional studies of *LAMP-2* expression patterns at the level of mRNA and proteins in various tumor tissues, as well as in embryonic tissues at various stages of development [84].

Our study showed the highest level of *LAMP-2* transcriptional activity in the clinical stage II of CRC, suggesting that the CMA process has a tumor-inhibitory effect in the later stages of the disease. The decrease in *LAMP-2* transcriptional activity in subsequent stages of cancer advancement indicates the inhibition of chaperone-dependent autophagy, which suggests the accumulation of damaged proteins in the cell, which may lead to tumor progression. Therefore, further research at the molecular level is needed in this field, which will help determine the relationship of the *LAMP-2* protein with cancer progression and identify signaling pathways contributing to the development of the disease.

*PINK1* (*PTEN-induced putative kinase 1*), unlike other genes studied, participates in the third type of autophagy, which is mitophagy, responsible for the removal of damaged mitochondria. This is a type of selective autophagy. Damaged mitochondria can be a signal of cell death, inflammation, or aging. Increased levels of dysfunctional mitochondria contribute to the pathogenesis of many diseases, including cancer [85,86,87,88,89]. *PINK1*, as well as E3 ubiquitin ligase Parkin, are defense mechanisms by which cancer cells resist mitochondrial apoptosis [89,90]. In contrast, defective mitophagy affects many cellular pathways characteristic of neoplastic tumors, and adequate mitochondrial clearance probably contributes to tumor suppression at multiple molecular levels [91]. Once mitochondrial damage occurs, cells initiate mitophagy to remove damaged mitochondria and even induce mitochondrial apoptosis resulting in cell death [92]. Many studies have confirmed that mitophagy has a double effect on the development of cancer, it can both induce and inhibit cancer progression [93]. Anticancer drugs cause stress in the body by inducing autophagy, which reduces the effectiveness of these drugs [94]. There are two signaling pathways for mitophagy, dependent and independent of ubiquitin. Ubiquitin-dependent mitophagy plays an important role in the elimination of dysfunctional organelles through various mitochondrial physiological properties [95]. It has been shown that multidrug-resistant cancer cells become more sensitive to chemotherapeutic drugs when ubiquitin-dependent mitophagy is inhibited [92]. Therefore, it is believed that the induction of apoptosis combined with the inhibition of mitophagy may be a potential treatment strategy for, for example, breast cancer [96]. *PINK1* in healthy cells is present at low levels; it accumulates during mitochondrial damage, increased mitochondrial ROS, depolarization, and accumulation of damaged proteins [97].

In summary, *PINK1* is a tumor suppressor that modulates cellular metabolism and promotes colorectal cancer cell death. *PINK1* inhibits CRC formation through metabolic reprogramming mediated by activating p53 and reducing acetyl-CoA production. These findings can potentially be used as therapeutic strategies in the treatment of CRC [98].

In our study, *PINK1* showed reduced expression only in SCII, while in other stages it showed overexpression. Therefore, the results are consistent with other results published in the literature, where it has been shown that *PINK1* may play a dual role in tumor progression. Loss of this protein reduces the expression of key genes involved in autophagy and inhibits the division of cancer cells, thus the development of the disease. In turn, overexpression induces proliferation, colony formation, migration, and invasive potential of tumor cells [93]. This is confirmed by the analysis of *PINK1* expression in the Human Protein Atlas, which states that *PINK1* may be either beneficial or detrimental depending on the type of cancer. For example, increased *PINK1* expression in liver, kidney, pancreatic and endometrial cancers was associated with improved overall survival, while breast, cervical, ovarian, lung, glioblastoma, and melanoma cancers were associated with a poor prognosis. According to data from The Cancer Genome Atlas (TCGA), high expression of *PINK1* was associated with a better prognosis for renal and uterine cancers. However, in the case of lung, esophageal, and ovarian cancers, low *PINK1* expression was associated with a better overall survival rate [99].

*PINK1* is a complex regulator in cancer etiology, regulated in part by the early and late stages of the disease pathogenesis. Summarizing the obtained results, *PINK1* can be used as a potential target in the treatment of cancer. Reducing the expression of this gene limits proliferation and inhibits cell division, and thus may be a direct blocker of the cell cycle in cancer. In addition, the inability of cells to divide in *PINK1* deficiency causes an increase in chromosomal aberrations, genetic instability, or aneuploidy, which can lead to the development of cancer.

Analyzing the literature data, *PINK1* may also be a potential biomarker to predict responses to treatment, especially in the case of therapies that are based on mitochondrial processes and autophagy. Further studies on *PINK1* and its function in tumor biology are urgently needed to better understand disease mechanisms and to determine the therapeutic effect of *PINK1* inhibition in cancer. The results of our study should stimulate additional research into *PINK1* as a potential master regulator of cellular metabolism, aging, and cancer.

The last gene tested was *FOXO1* (*Forkhead box O1*). It is a transcription factor that regulates the transcription of genes that play a role in tumor suppression, energy metabolism, lifespan extension, apoptosis, and resistance to oxidative stress [100,101,102,103]. Recent studies have shown that by regulating pro-apoptotic genes such *as BIM, FasL*, and *TRAIL*, *FOXOs* act as tumor suppressors, resulting in growth inhibition of many types of cancer, including prostate, breast, glioblastoma, and colorectal cancer [104,105,106,107].

The post-translational modification of proteins is important for various cellular processes such as cell regulation and development. There is more and more talk about *FOXO1* methylation. Chae et al. showed that insulin promotes *FOXO1* methylation, causing *FOXO1* degradation mediated by HMTase G9a activity [108]. In contrast, previous studies have shown that many types of cancer, such as colorectal, prostate, and breast cancer, were promoted by abnormal insulin production [109,110,111]. Chae et al. also found differences in *FOXO1* and *G9a* expression between healthy and colorectal cancer samples. Tissue matrix analysis showed lower levels of *FOXO1* expression in human colon cancer crypt cells compared to controls. The Human Protein Atlas database confirmed these results. It is hypothesized that the increase in crypt foci and colon cancer risk through insulin activity may be related to the induction of G9a, resulting in methylation-mediated degradation of *FOXO1*. A strong effect of *FOXO1* on colon cancer cell proliferation was also observed, strongly suggesting the role of G9a-mediated *FOXO1* methylation in CRC growth [109]. The Human Protein Atlas database has shown that the survival rate of patients with cancers of lower *FOXO1* expression is shorter than those of patients with cancers of higher *FOXO1* expression in colorectal, renal, and hepatic adenocarcinomas [112,113,114]. Chae et al. confirmed the results published earlier in their study. Overexpression of G9a and decreased expression of *FOXO1* was observed in a large group of patients. *FOXO1* expression was downregulated by G9a in most patients, especially the more advanced the disease. Together, this data suggests that G9a-mediated degradation of *FOXO1* may play an important role in colon cancer progression. To sum up, the results showed that *FOXO1* methylation is important for cell proliferation and colony formation in CRC cells. Studies suggest that the G9a inhibitor may have therapeutic potential in the treatment of CRC by inhibiting *FOXO1* degradation [108].

Our results are only partially consistent with the literature data, since in CSI, II, and III, we observe increased *FOXO1* expression, while in CSIV—decreased expression. The differences may be due to post-translational modifications of *FOXO1*, e.g., acetylation, which promotes autophagy, affecting the survival of cancer cells [115].

In turn, in other types of cancer, the increased transcriptional activity of FOXO1 has also been demonstrated, e.g., in squamous cell carcinoma of the esophagus [116] and breast cancer [117]. *FOXO1* may be a potential target for epigenetic diagnosis or treatment of cancer in the future and may serve as a potential marker for assessing cancer prognosis.

The main challenge in developing a therapy targeted at specific genes is investigating the molecular mechanisms of activation, inhibition, and function of these genes. Currently, several studies have been conducted in which the inhibition of autophagy enhances the response to radiotherapy, e.g., in patients with melanoma, esophageal cancer, and ovarian cancer [118,119]. However, thus far, autophagy inhibition alone is not an effective therapeutic method [120]. The efficacy of autophagy in promoting cell death has been demonstrated in many other cancer models such as breast cancer, leukemia, prostate cancer, and myeloma. However, to date, clinical trials have not conclusively demonstrated that the inhibition of autophagy associated with anti-cancer therapy provides patients with effective therapeutic benefits [120,121,122]. Currently, in oncology, protocols aimed at autophagy induction instead of inhibition are undergoing intensive study [35,37,121]. Nevertheless, to date, none of the currently licensed drugs have been developed to modulate autophagy, although some drugs do activate autophagy to some extent [123,124].

Traditional cancer treatment includes chemotherapy, surgery, and radiation therapy. To date, none of these treatments guarantee full recovery. Therefore, further research into innovative therapeutic strategies is needed. With the development of molecular biology techniques, additional markers for predicting treatment response and developing potential targeted therapies are becoming increasingly important.

Our preliminary findings on the high correlation between genes involved in autophagy and colorectal cancer suggest that the genes under study may be used as target therapy in the treatment of CRC.

The limitations of our study are the following:

There were only a small number of colorectal cancer specimens in each stage of clinical cancer—which may have influenced the results of the findings from the research. This might be a limitation of the study design, although it is possible that, even if a larger panel of specimens were analyzed, the observations would be supported. We will continue the study this topic in future research.

Intestinal specimens were taken using the classic technique of surgery—which includes both the epithelial and the stromal parts of the tissue—and by doing so, it is possible that there is greater variability in the genes analyzed, compared to taking biological material only from the epithelial part or only from the stromal part, also considering that in advanced tumors the amount of stromal tissue increases [125].

Only one reference gene *(beta-actin*) was used for endogenous control in the Colorectal Cancer RT-qPCR experiment. *β-actin* is commonly used to normalize molecular expression studies due to its high conservation as an endogenous housekeeping gene, but we are planning to use novel reference genes identified via NGS and “classical” reference genes [126] in our next study.

## 5. Conclusions

Autophagy plays a dual role in both tumor progression and suppression. Many studies conducted so far confirm the important role of aerophagy in carcinogenesis, both as a suppressor in the early stage of cancer and as a promoter in the later stage of cancer advancement.

This publication highlights the role of autophagy in oncogenesis, as well as the potential of autophagy as a therapeutic target in the treatment of colorectal adenocarcinoma.

The regulation of autophagy can be used as an effective intervention strategy in cancer prevention and therapy by preventing cancer development, limiting tumor progression, and increasing the effectiveness of treatment.

The standard methods of cancer treatment used so far are capable of significantly prolonging life and stopping the progression of the disease. However, a serious problem presented here is the progression of the tumor and the recurrence of the tumor after treatment, mainly due to the emergence of resistance to treatment. Autophagy certainly facilitates the survival of cancer cells in unfavorable environmental conditions. Therefore, in the near future, standard cancer treatments combined with the regulation of autophagy activity through activators or inhibitors can be considered as a potential anti-cancer therapy. However, further research is needed in this direction to understand how autophagy contributes to cancer development and treatment, and how the aerophagy pathway can be targeted and regulated during cancer prevention and treatment.

Molecular studies on the regulatory pathways of autophagy will certainly contribute to the development of effective preventative and therapeutic methods for colorectal cancer.

## Figures and Tables

**Figure 1 biomedicines-11-00418-f001:**
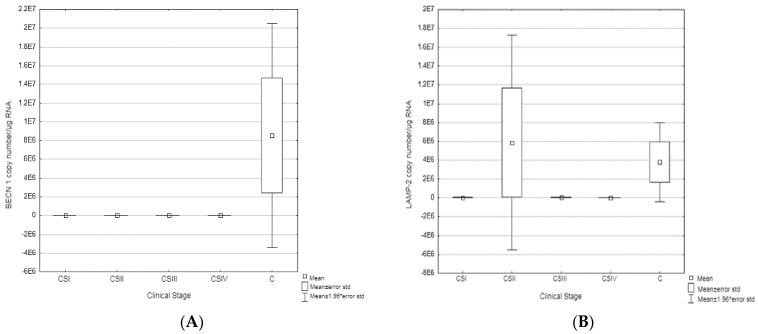
(**A**). Comparison of the number of *BECN1* mRNA molecules in clinical stages (I–IV) of CRC vs. controls (C). (**B**). Comparison of the number of *LAMP-2* mRNA molecules in clinical stages (I–IV) of CRC vs. controls (C).

**Figure 2 biomedicines-11-00418-f002:**
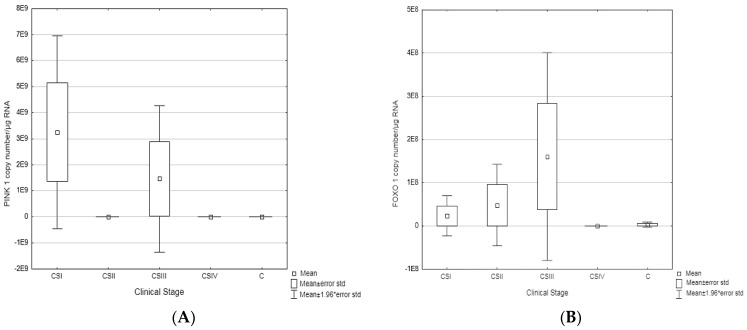
(**A**). Comparison of the number of PINK mRNA molecules in clinical stages (I–IV) of CRC vs. controls (C). (**B**). Comparison of the number of FOXO mRNA molecules in clinical stages (I–IV) of CRC vs. controls (C).

**Table 1 biomedicines-11-00418-t001:** Clinicopathological characteristics of the study group.

Feature	Value
Tumor location	Colon: 26 (66.67%); Rectum: 13 (33.33%)
T	I: 1 (2.56%); II: 16 (41.03%); III: 20 (51.28%); IV: 2 (5.13%)
N	N0: 24 (61.54%); N1: 4 (10.26%); N2: 11 (28.21%)
M	M0: 36 (92.31%); M1: 3 (7.69%)
TNM staging	I: 14 (35.9%); II: 9 (23.08%); III: 13 (33.33%); IV: 3 (7.69%)

**Table 2 biomedicines-11-00418-t002:** Sequences of the primers used for the RT-qPCR reaction.

Gene	Primer Sequences 5′ → 3′	
	Starter Forward	Starter Reverse
*BECN 1*	CAGTATCAGAGAGAATACAGTG	TGGAAGGTTGCATTAAAGAC
*LAMP-2*	AACAAAGAGCAGACTGTTTC	CAGCTGTAGAATACTTTCCTTG
*PINK 1*	GGACGCTGTTCCTCGTTA	ATCTGCGATCACCAGCCA
*FOXO 1*	GTCAAGACAACGACACATAG	AAACTAAAAGGGAGTTGGTG

**Table 3 biomedicines-11-00418-t003:** Values of descriptive statistics, ANOVA test and post hoc test of mRNA coding for the studied genes in colorectal cancer specimens and in the control specimens.

Colon Samples	Av	Me	Q1	Q3	SD	ANOVA Test	Post Hoc (LSD) Test
** *BECN1* **							
CSI	13,901	10,245.00	13,901	10,245.00	19,344	*p* = 0.759580	NS
CSII	9292	5981.00	9292	5981.00	13,083		
CSIII	12,258	6141.50	12,258	6141.50	17,626		
CSIV	4934	2629.00	4934	2629.00	5475		
C	8,530,219	10,560.00	8,530,219	10,560.00	57,606,226		
** *LAMP-2* **							
CSI	23,389	14,735.00	8417.00	23,340.00	25,031	*p* = 0.617352	NS
CSII	5,856,396	22,940.00	12,870.00	35,040.00	26,703,506		
CSIII	27,636	19,830.00	9676.00	47,850.00	22,862		
CSIV	17,467	17,775.00	14,455.00	20,895.00	4931		
C	3,786,237	33,850.00	14,310.00	51,660.00	20,487,421		
** *PINK1* **							
CSI	2248 × 10^9^	1405	2248 × 10^9^	1.405	8879 × 10^9^	*p* = 0.044194	CSII vs. CSI *p* = 0.024910 CSI vs. C *p* = 0.004188
CSII	1257 × 10^4^	1745.00	1257 × 10^4^	1745.00	3342 × 10^4^		
CSIII	1452 × 10^9^	153.00	1452 × 10^9^	153.00	8245 × 10^9^		
CSIV	3586 × 10^4^	7202.00	3586 × 10^4^	7202.00	7905 × 10^4^		
C	2285 × 10^4^	11.780	2285 × 10^4^	11.780	1809 × 10^5^		
** *FOXO1* **							
CSI	23,485,542	337.200	134.8500	3658.50	81,347,958	*p* = 0.219372	CSIII vs. C *p* = 0.020474
CSII	47,997,855	563.250	50.4000	5226.00	179,571,390		
CSIII	1,606,633,388	2143.500	48.1000	34,430.00	575,189,334		
CSIV	6128	275.500	74.3345	12,685.00	10,116		
C	3,116,668	262.800	1.3780	14,070.00	23,029,138		

Av—mathematical average; Me—median; Q1—lower quartile; Q3—upper quartile; SD—standard deviation; C—control group; CSI, CSII, CSIII, CSIV—clinical stages of colorectal cancer; NS—statistically insignificant difference; LSD—last significant differences test.

**Table 4 biomedicines-11-00418-t004:** Change in the expression of the studied genes determined by RT-qPCR in colorectal cancer and control (C).

Gene	Clinical Stage			
	CSI vs. C	CSII vs. C	CSIII vs. C	CSIV vs. C
*BECN1*	↓	↓	↓	↓
*LAMP-2*	↓	↑	↓	↓
*PINK1*	↑	↓	↑	↑
*FOXO1*	↑	↑	↑	↓

↑—increase in gene expression, ↓—decrease in gene expression.

## Data Availability

The data presented in this study are available on request from the corresponding author.

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
