# Peer review of "Transcription of Autophagy Associated Gene Expression as Possible Predictors of a Colorectal Cancer Prognosis"

_biomedicines, 2023, doi:10.3390/biomedicines11020418_

Round 1

Reviewer 1 Report

The authors analyzed the expression of four autophagy-associated genes in colorectal cancer samples. The analysis was performed on 39 colon cancer samples from men and women and in four different clinical stages. It increases the value of the collected samples for research. The authors described also the tumor location in colon. Control samples were from surgical margins. Only the differences between control and cancer sample in expression were significant for PINK1 gene. 

These results are strange. The most reasonable is the expression of BECN1 gene. Low expression in cancer tissue in comparison with control. The rest of analyses looks strange. LAMP2 – elevated expression in CSII and control, PINK1 (the only statistical significance) – elevated expression for CSI and CSIII, FOXO1 – elevated expression for CSI, CSII, CSIII and low expression for CSIV and control. It needs explanation especially for PINK1 gene where the differences in expression are statistically significant. 

The differences in the expression are extremely high. May be the crude results were not normalized before the statistical analysis.

Reviewer 2 Report

The authors recently published a paper in which they performed a microarray analysis (HGU 133A -
Affymetrix) of the expression profile of genes involved in autophagy (adenocarcinoma specimen Vs healthy
marginal surgical specimen as a control).

Now, in this article under review they aim to assess the changes in expression pattern of genes involved in
autophagy of colorectal cancer specimen. They moreover, removed surgical specimen form patients who
decided voluntarily in the participation of the study. Similarly, in this case histopathological surgical margin
tissue were taken as control.

Although, the experimental design was well organized and the genes studied are genes almost came from
the previous data published (I can see only the abstract, I don’t have a free access to the paper) and have a
high impact in autophagy, I think for the reason that you can see below the paper need a major revision.
The main reason that led me to suggest you a major review is that the primers of PINK1 don’t match with
the DNA template sequence in silico. In particular I used NCBI blast primer in order to see if the pair of
primers matched with the real target sequence of genes names declared. Especially for PINK1 reverse
primer, I tried to see if the match was true with the order sequence same as you write in the paper, also I
tried with the order sequence “reverse and complement”. In the first case the National Library Medicine
Database didn’t find the target sequence, instead, in second case it found others target sequence.

You have to find if you wrote wrong sequences of the couple pairs primers of PINK1 gene, or in other case
you have to repeat all experiment with PINK1 and all statistical analyses.

The second main reason, in order to have better and sure results, it would be necessary to use three
different reference genes, after a pre-running RT-qPCR with a multitude (ten for example) of reference
genes where you have to select the best three reference genes to use in your experiments. Therefore, you
need to repeat your experiments with three reference genes.

Minor revision:

1. I downloaded the datasheet of Qiagen kit that you used, it clear that you don’t follow their suggestion
protocols, did you validate your protocol before use with your patient’s cohort biopsy?

If no, this protocol needs to be validated, before start.

2. Can I see the standard deviation of your Beta-Actin Cts of your RT-qPCR experiments? Since, you used
TaqMan probes you should have limited the SD of the Cts of the experiments, but in general I wouldn’t use
the beta-Actin as endogenous control in samples with a high rate of proliferation between cancer cell and
normal colon cells. You might have a big SD between studies biopsy and control biopsy.

3. Table 4.

You should use the comma only for indicate the thousands, while if you use in the text for “p value
significance number the “point” (lane 166), in order to indicate decimal number, you should use the point
also for the “p value” number in the Table. Please change.

Moreover, in the legend of the table you indicate as “K” the control group data, while I can’t see the letter
K in the Table. You have to substitute the C (for the case control data) with K.

In case of FOXO1 you indicate as statistically significant the result Anova statistical test, but I can’t see
number below p<0.05. You inserted both for Anova and Post-Hoc test p = 0,219372 and p = 0,20474. Please
verify and change.

4. Figure 1-2. I printed the figures, but I can’t see in a proper way the titles of the axes, also the numbers of
the axes aren’t good for a proper analysis. Please increase the DPI of your figures.

5. Since you wrote in the lanes 146-149 that the obtained results did not meet the assumptions of normal
distribution, did you set the statistical program in order to get Anova with non-parametric results? In any
case you should add a sentence in lane 149 that say that you set the program for non-parametric test.

6. I saw in Figure 1 in the graph of BECN1 that the bar of control group was higher than CSI-CSIV data. Since,
from the Table 4 Anova statistical test did not produce any significant difference, did you set your statistical
program for multiple comparison? If no you should. Instead, if yes you should try to do statistical non-
parametric t-test between control group median and the median of every CSs. Similarly, you can try the same
method for data of LAMP2. In this case, beyond the difference with control group, I expected that you will
find difference between the cancer’s phases (CSI-CSIV).

7. Although, you cited several papers from high impact journals, you didn’t insert any reference from 2022,
and only three refs from 2021, and only four from 2020. You should insert a newer paper in your library,
especially because you insert 124 references.

The Reference n. 28 don’t appear in the text.

8. Please correct.

Lane 240 Beclina 1 in Beclin 1

Lane 241 Bekliny 1 in Beclin 1

Lane 252 Beklina 1 in Beclin 1

9. Table 2.

Leave the space between last word group and the final point.

Leave the space between Colon word and the double point.

Insert space after the double point of word Male and 18.

I hope I have made good suggestions for increasing the value of work.

Have a good work.

Round 2

Reviewer 2 Report

In this revision, authors have partially improved the completeness of reporting sections as well as the clarity and precision of their writing throughout the manuscript. 

Since they did not respond completely I have to reject the paper.

Here mine answers and questions for further upgrade of the work.

1. The authors have added the right sequence on forward and reverse primers of PINK1 gene, that now matching with the right PINK1 sequence (testing with primer BLAST online tool).

2. The authors did not perform a Medline search for best endogenous control in Colorectal Cancer RT-qPCR experiment, at the time of drawing the RT-qPCR protocol. (here an example of best endogenous control for CRC https://pubmed.ncbi.nlm.nih.gov/20122155/) 

They sentenced that “β actin is commonly used to normalize molecular expression studies due to its high conservation as an endogenous housekeeping gene”, and proposed three papers to support their thesis. When you need to design a real time PCR protocol, you must necessarily use stable endogenous controls, especially with tissues from different part of the gut, from different patients and a different degree of tumor staging.

Moreover, according to the authors, it is not possible at this point to use more endogenous controls to have a stronger data normalization (as suggested by https://www.gene-quantification.de/MIQE_qPCR&dPCR_5_eds_20022022.pdf).

I suggest to explain better the section “materials” where they described how they got the intestinal sections. I mean another problem to use β actin as endogenous control, is the multicellularity of the taken tissues. Did they use laser cutting before take the intestinal sections? Although they got tissues from different part of the gut, did they take always the epithelial or stromal section from the tumor biopsies?

3. I probably did not ask the question well; the authors suggest a real time PCR protocol that differs from those proposed by the Qiagen instructions. Did the authors university’s ethics committee intervene also in the scientific decisions? The ethics committee usually regulates the procedures for taking samples from patients and those relating to the purpose of using such samples.

4. I asked to the authors to show me the Cts of beta actin B endogenous control. It is not explained how required standard deviations were obtained. In my humble opinion if these were normal averages/medians of standard deviations obtained from calculation with DDct RT-method, would be very high number of standard deviations and this might suggest that in this work the authors have indirectly monitored how it changes the beta-actin B in their colorectal tissue samples.

I suggest to the authors to introduce in the aim of the introduction paragraph the use of method for RNA copy number variants calculation and reference/s. Moreover, in method section I would explain in more detail how the RNA copy number variants number was calculated, and I would add one or some references.

5. The new Table 3. The authors said that they changed “the comma” for the “p value”, but they forgot to change also the “p value” of PINK1 Post Hoc Test.

Moreover, add in the legend the acronym of LSD Post Hoc Test.

6. ok

7. ok

8. I printed again the pages where the authors inserted the Figure 1 and 2. Unfortunately the result is same as before, poor images quality.

Maybe the authors copy/transfer the graphs from their statistical program to power point in order to produce the final form of the figures. If yes, it is ok, but the authors need of another program to transfer the figure with the good quality to the final version of the manuscript in word. There are some free online.

Moreover, the authors must insert the data distribution over the bars, in order to help the readers in the comprehension of the graphs.

9. What doesn’t mean: “Because most of the data had a non-normal distribution a nonparametric test ANOVA Kruskala-Wallisa (p< 0.05) was used to assess the differences between the groups.”?

If some data were calculated with parametric test the authors need to insert.

Correct the polish version of the test in the English version (leave the finals “a”).

10. ok. But, if the authors use also the t test they could obtain more data to describe. Usually in the Anova the statistician leaves the outliers in the all groups. With single t-test the authors can add some data that with the Anova they left.

11. ok

12. lane 271 correct Beklina1 in Beclin1. Write LAMP2 same in the all text. Correct lane 313.

Change RNA LaterTM à in RNA LaterTM

SYBR Green it should always be written equal

Lane 460 add “.”

Lane 493 they cancel University of Silesia with Universi. What does mean?

Author Response

Manuscript:
Transcription of autophagy associated genes expression as possible predictors of colorectal
cancer prognosis

Martyna Bednarczyk, Małgorzata Muc-Wierzgoń
, Sylwia Dzięgielewska-Gęsiak, Edyta Fatyga
and Dariusz Waniczek

Reviewer’s opinion „ Since they did not respond completely I have to reject the paper”.

We do not agree with the above opinion because:

The presented review completely lacks any reference to the subject of the manuscript autophagy, autophagy gene expression of CRC, discussion of the results.

Reviewer (R ) stated that: The authors did not perform a Medline search for best endogenous control in Colorectal Cancer RT-qPCR experiment, at the time of drawing the RT-qPCR
protocol. (here an example of best endogenous control for
CRC
https://pubmed.ncbi.nlm.nih.gov/20122155/)

First: Reviewer refers primarily to the methodology of PCR analysis, even without considering that the results obtained using this method are similar to those described in the literature (see-
the section discussion of manuscript).
Research on the analysis of selected genes (not only autophagy genes) in CRC samples at
various stages of clinical advancement, has been conducted by the authors since 2015. The design of the PCR protocol that use stable endogenous controls, especially with tissues from
different part of the intestine, from different patients and a different degree of tumor staging was part of this research. Based on these studies/experiments, β-actin was used as an endogenous control in the present study - exemplary articles:

Expression profile of melatonin receptors and genes associated with their activity in
colorectal cancer. Małgorzata Muc-Wierzgoń et.al. Endocrinol Pol.2015,T 66, Supl.A

Evaluation of changes in transcriptional acitvity of a protein involved in the regulation of autophagy in different clinical stages of colon cancer. Bednarczyk M et al. In : Research and development of young scientists in Poland. Poznań, 2016; 65-72
Profile of Expression of Genes Encoding Matrix Metallopeptidase 9 (MMP9), Matrix Metallopeptidase 28 (MMP28) and TIMP Metallopeptidase Inhibitor 1 (TIMP1) in Colorectal Cancer: Assessment of the Role in Diagnosis and Prognostication.
Zbigniew Lorenc, Dariusz Waniczek et al. Med.Sci Monit. 2017 Mar 15;23:1305-1311

The transcriptional activity profile of inhibitor apoptosis protein encoding genes in colon cancer patients: A STROBE-compliant study. Dariusz Waniczek et.al Medicine
2021,
Nov 19;100(46):e27882.

Second: In the previous response we indicated three exemplary studies that allow the use of actin as the only regression gene. Reviewer as an example for the best endogenous control for
CRC gives us the article published in 2010 in BMC Mol Biol by
Elrasheid A H Kheirelseid , Kah Hoong Chang, John Newell, Michael J Kerin, Nicola Miller :Identification of endogenous
control genes for normalisation of real-time quantitative PCR data in colorectal cancer.

In study of Xu et al. ( Novel reference genes in colorectal cancer identify a distinct subset of high stage tumors and their associated histologically normal colonic tissues BMC Med Genet.
2019 Aug 13;20(1):138) the authors propose 8 novel gene identified via NGS with potentially unique biological properties as reference genes in CRC samples and pointing to the need for
further research of reference genes.

Therefore in limitations of our study we wrote - Limitation of our study are following:
1. Small number of colorectal cancer specimens in each stages of clinical cancer - which may
have influenced the results of the findings from the research. This might be a limitation of the
study design, although it is possible that, even if a larger panel of specimens were analyzed, the
observations would be supported. We will continue this topic in future researches.

2. Using one reference gene ( beta-actin) for endogenous control in Colorectal Cancer RT-
qPCR experiment.
β-actin is commonly used to normalize molecular expression studies due to
its high conservation as an endogenous
housekeeping gene, but we are planning to use in our
next study novel reference genes identified via NGS and “classical “ reference genes [124].

Reviewer (R ) stated that : I suggest to explain better the section “materials” where they described how they got the intestinal sections. I mean another problem to use β actin as
endogenous control, is the multicellularity of the taken tissues. Did they use laser cutting before take the intestinal sections? Although they got tissues from different part of the gut, did they
take always the epithelial or stromal section from the tumor biopsies?

Section “ Materials” has been completed. Actually ( in manuscript): Tissue samples were obtained during surgical resection of the colon affected by cancer, which was performed
according to surgical treatment standards. The tissue samples were collected using classical surgical techniques without the use of electric or ultrasound instruments. The material obtained
consisted of tumor tissue and/or healthy colon tissue. Healthy control tissue specimens were collected from an area 5 cm outside of the histologically negative margin, during the operation
because of CC. All materials were taken by the same operational team to minimize the mistakes.
The cancer samples were obtained from the margin of the resected material to rule out the presence of necrotic tissue in the specimens.

Reviewer (R ) stated that:
I probably did not ask the question well; the authors suggest a real time PCR protocol that differs from those proposed by the Qiagen instructions. Did the authors university’s ethics committee intervene also in the scientific decisions? The ethics committee usually regulates the procedures for taking samples from patients and those relating to the purpose of using such samples.

Supplemented information in the text of manuscript: The number of mRNA copies of the analyzed genes was determined based on the analysis of the kinetics of the QRT-PCR reaction
using the Engine OPTICONTM sequence detector (MJ Research) and the QuantiTectTM SYBRGreen RT-qPCR Kit (QIAGEN) reagents.
All according to the manufacturer's
recommendations [32].

Reviewer was unable to provide us in which points, we deviated from suggested Qiagen instructions.

Regarding second part of the question - yes, indeed. University’s ethics Committee range of competence is bit wider if researches touches particular branches like: taking biological samples
etc - Article 21(4) of the amended Act on the Profession of Physician and Dentist.

Reviewer (R ) stated that: I asked to the authors to show me the Cts of beta actin B endogenous control. It is not explained how required standard deviations were obtained. In my humble opinion if these were normal averages/medians of standard deviations obtained from calculation with DDct RT-method, would be very high number of standard deviations and this might suggest that in this work the authors have indirectly monitored how it changes the beta- actin B in their colorectal tissue samples.

The Cts of β-actin endogenous control:

CSI 18,49136

CSII 15,92667

CSIII 17,51455

CSIV 17,46083

C ( control group) 18,00578

Reviewer (R ) stated that: The new Table 3. The authors said that they changed “the comma” for the “p value”, but they forgot to change also the “p value” of PINK1 Post Hoc Test.

Moreover, add in the legend the acronym of LSD Post Hoc Test.

Corrected

Reviewer (R ) stated that: I printed again the pages where the authors inserted the Figure 1 and 2. Unfortunately the result is same as before, poor images quality.

Quality of print depends on quality and settings of printer.

Reviewer (R ) stated that Maybe the authors copy/transfer the graphs from their statistical program to power point in order to produce the final form of the figures. If yes, it is ok, but the
authors need of another program to transfer the figure with the good quality to the final version of the manuscript in word. There are some free online.

Moreover, the authors must insert the data distribution over the bars, in order to help the readers in the comprehension of the graphs.

We always use another program for quality improvement of our figures. Usually more complex
quality improvement and resizing/resampling are left for “preparation for printing” stage.

Reviewer (R ) stated that : What doesn’t mean: “Because most of the data had a non-normal distribution a nonparametric test ANOVA Kruskal-Wallis (p< 0.05) was used to assess the
differences between the groups.”? If some data were calculated with parametric test the authors need to insert.
In text of manuscript there is written: 2.3.1 Statistical analysis

The results obtained by using the qRT-PCR technique were developed based on the
the Statistica 12.5 program (StatSoft, Tulsa, OK, USA). For each analyzed parameter, the most
important elements of descriptive statistics were determined: mean, median, minimum and maximum value, standard deviation, and upper (75%) and lower (25%) quartiles.
The normality
of value distribution was checked by Shapiro-Wilk test. Then, the results with a Gaussian distribution were analyzed with Student’s -test, and those with a non-Gaussian distribution were
verified by a nonparametric Kruskal-Wallis test to assess the differences between studied groups. A p<0.05 was taken as indicative of significant differences. The results are expressed
as mean and standard deviation (SD).

In this revision, authors have partially improved the completeness of reporting sections as wellas the clarity and precision of their writing throughout the manuscript

Yours Sincerely,

Małgorzata Muc-Wierzgoń

Round 3

Reviewer 2 Report

Revision 2 of the biomedicines 2123088 (answer in red) 

see the attached file for the images

Manuscript:

Transcription of autophagy – associated genes expression as possible predictors of colorectal
cancer prognosis
Martyna Bednarczyk, MaÅ‚gorzata Muc-WierzgoÅ„, Sylwia DziÄ™gielewska-GÄ™siak, Edyta Fatyga
and Dariusz Waniczek

Reviewer’s opinion, since they did not respond completely I have to reject the paper”.

We do not agree with the above opinion because:
The presented review completely lacks any reference to the subject of the manuscript –autophagy, autophagy gene expression of CRC, discussion of the results.

Because those parts are good.

Reviewer (R) stated that: The authors did not perform a Medline search for best endogenous control in Colorectal Cancer RT-qPCR experiment, at the time of drawing the RT-qPCR
protocol. (here an example of best endogenous control for
CRC https://pubmed.ncbi.nlm.nih.gov/20122155/)

First: Reviewer refers primarily to the methodology of PCR analysis, even without considering that the results obtained using this method are similar to those described in the literature (see-
the section “discussion of manuscript”). Ok, I focused on the technical part because it is a manuscript with only one analysis technique, so I think it must be done extremely well in order to give to the scientific community extremely valid data.

Research on the analysis of selected genes (not only autophagy genes) in CRC samples at various stages of clinical advancement, has been conducted by the authors since 2015. 

The design of the PCR protocol that use stable endogenous controls, especially with tissues from
different part of the intestine, from different patients and a different degree of tumor staging was part of this research. 

Based on these studies/experiments, β-actin was used as an endogenous control in the present study - exemplary articles:

• Expression profile of melatonin receptors and genes associated with their activity in
colorectal cancer. MaÅ‚gorzata Muc-WierzgoÅ„ et.al. Endocrinol Pol. 2015,T 66, Supl.A

• Evaluation of changes in transcriptional acitvity of a protein involved in the regulation of autophagy in different clinical stages of colon cancer. Bednarczyk M et al. In: Research and development of young scientists in Poland. PoznaÅ„, 2016; 65-72

• Profile of Expression of Genes Encoding Matrix Metallopeptidase 9 (MMP9), Matrix Metallopeptidase 28 (MMP28) and TIMP Metallopeptidase Inhibitor 1 (TIMP1) in Colorectal Cancer: Assessment of the Role in Diagnosis and Prognostication.
Zbigniew Lorenc, Dariusz Waniczek et al. Med.Sci Monit. 2017 Mar 15;23:1305-1311

• The transcriptional activity profile of inhibitor apoptosis protein encoding genes in colon cancer patients: A STROBE-compliant study. Dariusz Waniczek et.al Medicine
2021, Nov 19;100(46):e27882.

Although, the first of the four copied references is not online but there is only one published letter to the editor derived from the work itself and not even the second there is no trace, after my first review I would have preferred these works of your group instead of some random work where the authors used beta actin as an endogenous control but not on tissues derived from the colon.

Second: In the previous response we indicated three exemplary studies that allow the use of actin as the only regression gene. Reviewer as an example for the best endogenous control for
CRC gives us the article published in 2010 in BMC Mol Biol by Elrasheid A H Kheirelseid , Kah Hoong Chang, John Newell, Michael J Kerin, Nicola Miller: Identification of endogenous
control genes for normalization of real-time quantitative PCR data in colorectal cancer.

Of course, that’s what I would do. Although this takes more time and also more cost.

In study of Xu et al. (Novel reference genes in colorectal cancer identify a distinct subset of high stage tumors and their associated histologically normal colonic tissues BMC Med Genet.
2019 Aug 13;20(1):138) the authors propose 8 novel gene identified via NGS with potentially unique biological properties as reference genes in CRC samples and pointing to the need for
further research of reference genes.

Also this paper confirms my doubts.

Therefore, in limitations of our study we wrote – 

Limitation of our study are following:

1. Small number of colorectal cancer specimens in each stages of clinical cancer - which may
have influenced the results of the findings from the research. This might be a limitation of the
study design, although it is possible that, even if a larger panel of specimens were analyzed, the
observations would be supported. We will continue this topic in future researches.

I would add that with the classic technique of surgery you have taken both the epithelial and the stromal part of the tissue and by doing so it is possible that there is greater variability of the genes you have studied; compared to taking in all the intestinal sites (which you have taken) always the epithelial part or always the stromal part. Considering also that in advanced tumors increases the part of stromal tissue.

I leave you a support reference

https://www.ncbi.nlm.nih.gov/pmc/articles/PMC9139914/

2. Using one reference gene (beta-actin) for endogenous control in Colorectal Cancer RT-
qPCR experiment. 
β-actin is commonly used to normalize molecular expression studies due to
its high conservation as an endogenous housekeeping gene, but we are planning to use in our
next study novel reference genes identified via NGS and “classical “reference genes [124].

ok

Reviewer (R) stated that: I suggest to explain better the section “materials” where they described how they got the intestinal sections. I mean another problem to use β actin as
endogenous control, is the multicellularity of the taken tissues. Did they use laser cutting before take the intestinal sections? Although they got tissues from different part of the gut, did they
take always the epithelial or stromal section from the tumor biopsies?

Section “Materials” has been completed. Actually (in manuscript): 

Tissue samples were obtained during surgical resection of the colon affected by cancer, which was performed according to surgical treatment standards. The tissue samples were collected using classical surgical techniques without the use of electric or ultrasound instruments. The material obtained
consisted of tumor tissue and/or healthy colon tissue. Healthy control tissue specimens were collected from an area 5 cm outside of the histologically negative margin, during the operation
because of CC. All materials were taken by the same operational team to minimize the mistakes.
The cancer samples were obtained from the margin of the resected material to rule out the presence of necrotic tissue in the specimens.

Ok, now it is better and more understandable.

Reviewer (R) stated that: I probably did not ask the question well; the authors suggest a real time PCR protocol that differs from those proposed by the Qiagen instructions. Did the authors university’s ethics committee intervene also in the scientific decisions? The ethics committee usually regulates the procedures for taking samples from patients and those relating to the purpose of using such samples.

Supplemented information in the text of manuscript: The number of mRNA copies of the analyzed genes was determined based on the analysis of the kinetics of the QRT-PCR reaction
using the Engine OPTICONTM sequence detector (MJ Research) and the QuantiTectTM SYBRGreen RT-qPCR Kit (QIAGEN) reagents. All according to the manufacturer's recommendations [32].
Reviewer was unable to provide us in which points, we deviated from suggested Qiagen instructions.

I leave you some pictures of my screen.

Your running qPCR protocol (What you have written)

“The thermal profile of the RT-qPCR reaction consisted of the following steps: reverse 153 transcription (45ºC for 10 minutes), polymerase activation (95ºC for 2 minutes), 40 cycles 154 of denaturation (95ºC for 5 seconds), primer annealing (60ºC for 10 seconds), elongation 155 (72ºC for 5 seconds). The reaction was performed using sequence-specific primer pairs for 156 each gene tested (Sigma-Aldrich, St Louis, MO, USA);”

I downloaded the QuantiTect SYBR Green RT-PCR datasheet (name: QuantiTect SYBR Green RT-PCR Handbook) of the kit that you used. Within the manual there are qPCR running protocols that are different from yours. I attached you the photo of QuantiTect SYBR Green RT-PCR Handbook running qPCR protocols:

1.

2.

As you can see the QuantiTect SYBR Green RT-PCR Handbook qPCR running protocols are different than yours.

So, I’m asking again: did you validate your protocol before use with your patient’s cohort biopsy? If yes, please explain. It would also be fine if you did it for the previous papers that you sent in this review, if you used the same kit.

For me, the answer about the ethics committee is not enough; ok for taking biological samples, but is not enough because they do not have molecular biologists in their offices.

You have to change the reference 32 with the “QuantiTect SYBR Green RT-PCR Handbook” from Qiagen. The reference 32 means nothing.

Regarding second part of the question - yes, indeed. University’s ethics Committee range of competence is bit wider if researches touches particular branches like: taking biological samples
etc. - Article 21(4) of the amended Act on the Profession of Physician and Dentist.

Read the answers above.

Reviewer (R) stated that: I asked to the authors to show me the Cts of beta actin B endogenous control. It is not explained how required standard deviations were obtained. In my humble opinion if these were normal averages/medians of standard deviations obtained from calculation with DDct RT-method, would be very high number of standard deviations and this might suggest that in this work the authors have indirectly monitored how it changes the beta- actin B in their colorectal tissue samples.

The Cts of β-actin endogenous control:
CSI 18,49136
CSII 15,92667
CSIII 17,51455
CSIV 17,46083
C (control group) 18,00578

Ok, this provide that the beta actin B is stable among the various CS and also in the control group.

What did the numbers you sent me in the last review represent?

I leave a copy here of your numbers:

“The standard deviation for B-actin:

CSI: 4,611456

CSII: 6,246788

CSIII: 6,098318

CSIV: 5,086564

Control: 7,013376”

you wrote “standard deviation”.

What does that mean? Means perhaps, that the average of the ct of the beta actin B of your samples, in the various groups, has the above listed standard deviation?   

Reviewer (R) stated that: The new Table 3. The authors said that they changed “the comma” for the “p value”, but they forgot to change also the “p value” of PINK1 Post Hoc Test.
Moreover, add in the legend the acronym of LSD Post Hoc Test.

Corrected

Well done.

Reviewer (R) stated that: I printed again the pages where the authors inserted the Figure 1 and 2. Unfortunately the result is same as before, poor images quality.

Quality of print depends on quality and settings of printer.

Let’s see if we understand each other now: I’m sorry but the images you provided are terrible even on video. If I increase the zoom % the captions in the lower right become blur, they are not sharp.

You need to find a solution.

Reviewer (R) stated that Maybe the authors copy/transfer the graphs from their statistical program to power point in order to produce the final form of the figures. If yes, it is ok, but the
authors need of another program to transfer the figure with the good quality to the final version of the manuscript in word. There are some free online.
Moreover, the authors must insert the data distribution over the bars, in order to help the readers in the comprehension of the graphs.

We always use another program for quality improvement of our figures. Usually more complex
quality improvement and resizing/resampling are left for “preparation for printing” stage.

Provide the images you would send for the press, I do not accept these, I’m sorry.

Moreover, the authors must insert the data distribution over the graph bars, in order to help the readers in the comprehension of the graphs.

Reviewer (R) stated that: What doesn’t mean: “Because most of the data had a non-normal distribution a nonparametric test ANOVA Kruskal-Wallis (p< 0.05) was used to assess the differences between the groups.”? If some data were calculated with parametric test the authors need to insert.

I mean, if you write “most of the data” in the text, means that some data are calculated in a different way. Thus, you have to write in the text. You have to be clear for the readers, otherwise it seems that you want to hide something, and I don’t think it’s your case.

In text of manuscript there is written: 2.3.1 Statistical analysis
The results obtained by using the qRT-PCR technique were developed based on the Statistica 12.5 program (StatSoft, Tulsa, OK, USA).

For each analyzed parameter, the most important elements of descriptive statistics were determined: mean, median, minimum and maximum value, standard deviation, and upper (75%) and lower (25%) quartiles. 

The normality of value distribution was checked by Shapiro-Wilk test. Then, the results with a Gaussian distribution were analyzed with Student’s -test, and those with a non-Gaussian distribution were
verified by a nonparametric Kruskal-Wallis test to assess the differences between studied groups. A p<0.05 was taken as indicative of significant differences. The results are expressed
as mean and standard deviation (SD).

if you have used the Kruskal-Wallis test (for non-Gaussian distribution data), it uses the median to verify the equality between groups, therefore the results should be shown as median and standard deviation.

Please change in the main text, I would write: in case of Gaussian distribution the results are expressed as mean and standard deviation (SD), instead in case of non-Gaussian distribution the results are expressed as median and standard deviation.

In this revision, authors have partially improved the completeness of reporting sections as well as the clarity and precision of their writing throughout the manuscript

As I wrote you partially answered to my suggestion.

You forgot to change:

Correct lane 274 from Polish name Beklina1 to English name Beclin 1.

Correct lane 262 Beclin1 à add space as other name Beclin 1

Write LAMP2 same in the all text. Correct lane 316 you wrote LAMP-2.

Correct lane 139 Change RNA LaterTM à in RNA Later TM

Correct lane 138 OPTICOMTM à OPTICOM TM

Correct lane 139 QuantiTect TM à QuantiTect TM

Write RT-qPCR same in the text, you wrote in four different methods, see lane 138, 139, 158, 162. Choose one and use that.

It was not an order, as written here, I report as I had written in the previous revision:

“I suggest to the authors to introduce in the aim of the introduction paragraph the use of method for RNA copy number variants calculation and reference/s. Moreover, in method section I would explain in more detail how the RNA copy number variants number was calculated, and I would add one or some references.”

I’ll tell you again: “when you explained in the methods how you got RNA copy numbers, I think you need to expand and explain this part better.” From lane 146 to lane 151.
If you are unable to deepen this section, you could add one reference that explain how RNA copy numbers is obtained.

Yours Sincerely,
Małgorzata Muc-Wierzgoń

I hope I made some improvements to the paper, good job.

Author Response

Please see the attechment

Round 4

Reviewer 2 Report

Dear Authors, (answer in red)

I really appreciate the efforts you are making to improve your work according to my advices.

But I’m awfully sorry to tell you that I have to reject the work again with minor revision.

Unfortunately, there are still inconsistencies about what you wrote in the text and in previous revisions, and what you added this time.

I try to explain it close to your answers.

Dear Reviewer,

Transcription of autophagy – associated genes expression as possible predictors of colorectal cancer prognosis

Thank you very much for your comments regarding manuscript:

Martyna Bednarczyk , Małgorzata Muc-Wierzgoń , Sylwia Dzięgielewska-Gęsiak , Edyta Fatyga and Dariusz Waniczek

We modified our manuscript according to the enclosed Reviewers’ reports. I made some

corrections. The changes are presented in red color in manuscript

Answer

I would have preferred these works of your group instead of some random work where the authors used beta actin as an endogenous control but not on tissues derived from the colon

Unfortunately, these are works that were published as chapters in Polish monographs and they were not indexed in known medical databases. They have International Standard Book Number (ISBN):

no problems.

                     ï‚·ZmarzÅ‚y Nikola, Grabarek Beniamin, Wojdas Emilia, Bednarczyk Martyna, Kaźmierczak Agata, Gola Joanna, Asman Marek, Solarz Krzysztof, Strzelczyk Joanna Katarzyna, Zalewska-Ziob Marzena, WielkoszyÅ„ski Tomasz, Kurek Józef, KoleżyÅ„ska Barbara, Skubis Aleksandra, Mazurek Urszula Expression profile of genes associated with the activity of SMAD proteins in patients with Lyme disease : Arthropods. In urban and suburbian environments. Red. Alicji Buczek, CzesÅ‚awa BÅ‚aszaka Lublin : Koliber, 2017 p.101-110 p-ISBN: 978-83-60497-28-9

                     ï‚·Grabarek Beniamin, ZmarzÅ‚y Nikola, Wojdas Emilia, Bednarczyk Martyna, Kaźmierczak Agata, Gola Joanna, Asman Marek, Solarz Krzysztof, Strzelczyk Joanna Katarzyna, Zalewska-Ziob Marzena, WielkoszyÅ„ski Tomasz, Kurek Józef, KoleżyÅ„ska Barbara, Sikora Bartosz, Mazurek Urszula The influence of borrelia sp. infection on transcriptional activity of genes associated with JAK/STAT signal pathway activated by interleukin 12 and 23. Arthropods. In urban and suburbian environments.

Red. Alicji Buczek, Czesława Błaszaka Lublin : Koliber, 2017,p.133-142 p-ISBN: 978-83-60497-28-9

                     ï‚·Bednarczyk Martyna, ZmarzÅ‚y Nikola, Grabarek Beniamin, Wojdas Emilia, Gola Joanna, Asman Marek, Solarz Krzysztof, Strzelczyk Joanna Katarzyna, Zalewska-Ziob Marzena, WielkoszyÅ„ski Tomasz, Kurek Józef, KoleżyÅ„ska Barbara, Walkiewicz Katarzyna, Mazurek Urszula, Muc-WierzgoÅ„ MaÅ‚gorzata Expression profile of gene related of ubiquitin and autophagy in patients with Lyme disease. Arthropods. In urban and suburbian environments.

Red. Alicji Buczek, Czesława Błaszak Lublin : Koliber, 2017, p.111-122 p-ISBN: 978-83-60497-28-9

                     ï‚·Bednarczyk Martyna, ZmarzÅ‚y Nikola, Kaźmierczak Agata, Grabarek Beniamin, Dudek SÅ‚awomir, Mazurek Urszula M., Muc-WierzgoÅ„ MaÅ‚gorzata Detection profile of chaperone genes infibroblasts cells of Borrelia burgdorferi. Arthropods at the beginning of the new century. Red. Alicji Buczek, CzesÅ‚awa BÅ‚aszaka. Lublin : Koliber, 2018 p.75-83 p-ISBN: 978-83-60497-38-8

Limitations of our study (supplemented according your suggestions): well done

1. Small number of colorectal cancer specimens in each stages of clinical cancer – it may have influenced the results of the findings from the research. This might be a limitation of the study design, although it is possible that, even if a larger panel of specimens were analyzed, our observations would be supported. We plan to continue this topic in future researches.

2. Intestinal specimens taken by classic technique of surgery - this is both the epithelial and the stromal part of the tissue and by doing so it is possible that there is greater variability of the genes analyzed; compared to taking biological material always from the epithelial part or always from the stromal part. Considering also that in advanced tumors increases the part of stromal tissue [125]

3. Using one reference gene (beta-actin) for endogenous control in Colorectal Cancer RT-qPCR experiment. β-actin is commonly used to normalize molecular expression studies due to its high conservation as an endogenous housekeeping gene, currently we

are planning to use in our next study novel reference genes identified via NGS and “classical “ reference genes [126].

I downloaded the QuantiTect SYBR Green RT-PCR datasheet (name: QuantiTect SYBR Green RT-PCR Handbook) of the kit that you used. Within the manual there are qPCR running protocols that are different from yours….

I am very sorry, in the second round of answer in order to complete the data, the parameters of analysis of the kinetics of the QRT-PCR reaction using the SensiFAST™ SYBR® No-ROX One-Step were given – print screen (https://dnagdansk.com/wp-content/uploads/2021/05/pi-50221_sensifast_sybr_no-rox_one-step_kit_a3_v12.pdf).

An article on diabetology is currently being prepared where SensiFAST™ SYBR® No-ROX One-Step were used.

In manuscript is: The number of mRNA copies of the analyzed genes was determined based on the analysis of the kinetics of the QRT-PCR reaction using the Engine OPTICONTM sequence detector (MJ Research) and the QuantiTectTM SYBRGreen RT-qPCR Kit (QIAGEN) reagents. according to the manufacturer's recommendations [32].

Why you have written that you use “QuantiTectTM SYBRGreen RT-qPCR Kit (QIAGEN)” and now you wrote that you used, instead, SensiFAST™ SYBR® No-ROX One-Step kit?

However, change in the text that you use SensiFAST™ SYBR® No-ROX One-Step instead QuantiTectTM SYBRGreen RT-qPCR Kit (QIAGEN). Change the reference 32 (again) with “https://dnagdansk.com/wp-content/uploads/2021/05/pi-50221_sensifast_sybr_no-rox_one-step_kit_a3_v12.pdf).”

Beta-actin problems..

Yes, that the average of the CT of the beta actin B of samples, in the various groups, has the above listed standard deviation.

Although your results are in line with those of the literature, this answer might leads to a true methodological problem and to the interpretation of the results.

I try to explain:

I reported the data that you sent to me:

The Cts of β-actin endogenous control:

CSI 18,49136
CSII 15,92667
CSIII 17,51455
CSIV 17,46083
C (control group) 18,00578

“The standard deviation for B-actin:

CSI: 4,611456

CSII: 6,246788

CSIII: 6,098318

CSIV: 5,086564

Control: 7,013376”

These data tell us, given the standard deviations that you have monitored, that the Cts of your beta actin samples in the various groups are very different from sample to sample and between the various pieces of intestines from which you extracted RNA.

Thus, here is the table of your beta actin Cts, mean (min; max):

Group

Min (Ct)

Mean (Ct)

Max (Ct)

CSI

13,879904

18,49136

23,102816

CSII

9,679882

15,92667

22,173458

CSIII

11,416232

17,51455

23,612868

CSIV

12,374266

17,46083

22,547394

Control

10,992404

18,00578

25,019156

I want to be explicit and help you publish this paper.

If I were to use this data of Cts of beta actin for the classic DDCt method, it is obvious that the standard deviation in the various groups is too high and would not be good for the relative calculation of an RT-qPCR. If the experiments were my own, I’d accept a standard deviation of no more than 2 per group.

I think I expressed myself badly in previous revisions, show me the standard deviation of the points of the calibration curve of the beta actin of the various groups.

I printed again the pages where the authors inserted the Figure 1 and 2. Unfortunately the result is same as before, poor images quality.

A new version of Figures has been prepared in the program: Pixlr X and Pixlr E.

The Images for me are better now. I leave it to the Editor to decide whether or not they meet their standards.

In previous revision I asked you this: “Moreover, the authors must insert the data distribution over the graph bars, in order to help the readers in the comprehension of the graphs.”

In the Images that you changed in the revised paper I don’t see what I asked above; I tried to ask you in a different way:

I want that you change again your figures with scatter plot over the bar.

See the Image below, or in the attached files.

Statistical analysis

According to your suggestions we corrected in main text:

The results obtained by using the qRT-PCR technique were developed based on the Statistica 12.5 program (StatSoft, Tulsa, OK, USA).

For each analyzed parameter, the most important elements of descriptive statistics were determined: mean, median, minimum and maximum value, standard deviation, and upper (75%) and lower (25%) quartiles. The normality of value distribution was checked by Shapiro-Wilk test. Then, the results with a Gaussian distribution were analyzed with Student’s -test. In case of Gaussian distribution, the results are expressed as mean and standard deviation (SD), instead in case of non-Gaussian distribution the results are expressed as median and standard deviation.

Ok, well done.

I’ll tell you again: “when you explained in the methods how you got RNA copy numbers, I think you need to expand and explain this part better.” From lane 146 to lane 151. If you are unable to deepen this section, you could add one reference that explain how RNA copy numbers is obtained.

Supplemented in main text: Simultaneously with the test samples, the amplification reaction of a commercially available quantitative standard - a fragment of the β actin gene (TaqMan® DNA Template Reagents Kit and β-actin Control Reagent Kit - Applied Biosystems) was performed, based on which a standard curve was determined, which is the basis for calculating the number of mRNA copies of the tested genes [33].

[33] - Guide to Performing Relative Quantitation of Gene Expression Using Real-Time Quantitative PCR. Applied Biosystems 2004, p.1-60

I search on this reference that you add for the method to calculate “RNA copy number”. It is possible that I didn’t find the method. Could you provide a screen shoot where the method is explained?

If inside this reference they don’t explain how calculate the RNA copy number, please insert a clearer (for the readers) reference.

Additionaly, corrected in main text: Beclin 1, LAMP-2, RT-qPCR

Remain to change QRT-PCR to qRT-PCR in the lanes 124-125

Sincerely,

Malgorzata Muc-Wierzgoń

I hope I made some improvements to the paper, good job.
